# Game and Simulation Stimulate Conceptual Change about Molecular Emergence in Different Ways, with Potential Cultural Implications

Andrea Gauthier [1,2]

1   Biomedical Communications, Institute of Medical Science, University of Toronto,
    Toronto, ON M5S 1A1, Canada; andrea.gauthier@ucl.ac.uk
2   UCL Knowledge Lab, IOE Faculty of Education and Society, University College London,
    London WC1E 6BT, UK

**Abstract:** Many undergraduate students hold robust misconceptions about the emergent nature of molecular processes, believing them to be directed rather than random. Interactive simulations might help transform such misconceptions by visualizing stochastic processes in a time-independent medium and enabling students to manipulate the environment to test their naïve hypotheses. Furthermore, game-based learning (GBL) might enhance the effectiveness of such simulations by promoting productive negativity (PN), i.e., learning from failure. In a randomized controlled trial with pre-test, post-test and delayed one-year follow-up, undergraduates ($n = 84$) engaged with either a GBL or interactive simulation (SIM) environment for 20–45 min and were compared to a baseline group ($n = 138$). GBL ($p = 0.035$) and SIM ($p = 0.069$) resolved more misconceptions than baseline but did not differ from each other ($p = 0.992$). GBL group also trended toward more positive long-term conceptual change. In-game interactions generated in response to PN were predictive of conceptual change in the GBL group alone, suggesting that PN may only be effective when supported by game design. Participants' native English-speaking status had a moderating effect, with native-speakers performing well in GBL and poorly in SIM environment, while the opposite was true for non-native-speakers, which, as discussed herein, may be aligned with cultural differences in acceptability of GBL. The GBL intervention generated longer voluntary use ($p = 0.005$), especially amongst frequent game-players. The results inform how GBL/SIM approaches can implement PN as a mechanism for conceptual change about molecular emergence.

**Keywords:** molecular randomness; misconceptions; conceptual change; productive negativity; game-based learning; simulations; culture

## 1. Introduction

Undergraduate molecular biology students bring with them a myriad of scientific misconceptions reinforced by their prior experiences in the world. A particularly problematic misconception is the idea that molecules have some sort of innate agency that guides their interactions in the cell, which conflicts with their post-secondary instruction about how molecular processes emerge due to the random or stochastic motion of all constituent parts [1–5]. It is thought that such misconceptions arise because students lack real-world referents for emergent processes, as they often occur at a spatial and temporal scales that are not easily observable. As such, these newly taught concepts about random molecular motion are assimilated with students' previous knowledge, and the misconception persists [6].

There is some concern that we may be reinforcing these misconceptions with the visual learning materials we provide in biology education. Animations and illustrations for molecular biology instruction typically visualize emergent molecular processes as linear, with steps occurring in a distinct sequence [7,8]. There are both pedagogical reasons for

this (i.e., to show the steps of the process clearly, so they can be learned), and reasons based on the limitations of these media (e.g., static illustrations requiring the use of arrows, and dynamic animations requiring the story to be told in a short period of time). As such, there is a risk of reinforcing students' misconceptions of molecular emergence as sequential and directed [7,8]. Whilst there is some promise of preventing the formation of misconceptions through animations that depict the dynamic, stochastic motion of molecules [9–11], there is a lack of evidence that such animations help students resolve existing misconceptions.

In contrast to illustrations and animations, interactive molecular simulations are both dynamic and time independent. They might support a transformative experience by serving as a visual model or exemplar of an emergent system [4,12], so that students can build a mental model of emergence through manipulating the simulations and testing their naïve hypotheses about molecular motion. The author also postulates that game-based learning (GBL) approaches might enhance the effectiveness of such simulations for conceptual change by (i) providing opportunities for "productive negativity" (i.e., learning from failure) and (ii) fostering increased motivation necessary for conceptual change to take place.

This study builds on previous research investigating how game design influences interactions in, and the efficacy of, an interactive simulation (SIM) designed to facilitate conceptual change about molecular emergence. It compares the effectiveness of two highly similar GBL and SIM interventions in contrast to standard instruction on both immediate and long-term resolution of misconceptions, in line with work touting the importance of making value-added comparisons to investigate media design [13,14]. While this study confirms the benefit of game design for conceptual change found in the author's previous work, it contributes novel findings related to the retention of conceptual change over time, as well as the mediating effect of student characteristics on the effectiveness of the intervention, which have been highlighted as long-standing gaps in the GBL literature [14–16]. The findings contribute to the discourse on how to better support students' formation of robust mental models of complex molecular biology concepts through interactive media.

### 1.1. Misconceptions and Conceptual Change

The problem of naïve misconceptions in life sciences has been recognized for many decades, and the learning science surrounding conceptual change in this domain continues to evolve. Misconceptions are established in learners when new knowledge conflicts with prior beliefs, so the new knowledge is either rejected altogether or is distorted in some way to assimilate within existing mental models to avoid cognitive conflict [17,18]. Commonly accepted conditions under which misconceptions may be transformed have long been proposed [19]: the learner must (a) be dissatisfied with their existing conception and must (b) find the new or alternative conception to be intelligible, (c) plausible, and (d) fruitful, meaning that they understand the concept and how it can be more broadly adapted.

Over the past decade, we have begun to understand that different types of misconceptions require different conceptual change strategies; Chi [6] describes some misconceptions as being inaccurate, while others are incommensurate in relation to learners' ontologically organized mental models. False beliefs (i.e., incorrect facts) and flawed mental models (i.e., several incorrect facts that make up a mental model) constitute the inaccurate type of misconceptions that can be relatively easily resolved through direct refutation of the false beliefs or through holistic mental model confrontation [6], as long as the conditions set forth by Posner et al. [19], listed above, are satisfied. Contrastingly, incommensurate types of misconceptions involve ideas that have been ontologically miscategorized. A simple example given by Chi [6] regards children's naïve belief that whales are a type of fish; since many children have a preconceived mental model of mammals, they can make the categorical shift from believing whales to be fish to considering them as mammals without much challenge, after being made aware of their miscategorization. However, if the learner does not already hold a metal model for the ontological category on which the misconceived concept needs to be shifted, conceptual change cannot occur [2,4,20–22].

Firstly, the learner must recognize that their conception is incorrect [6,19–21,23,24]; and secondly, they must construct a new ontologically accurate mental model to house the new conception [6,20,21].

This is often the case with emergent type processes (e.g., heat, force, diffusion), which are regularly misconceived as either sequential processes or as entities, like substances (e.g., ice cubes contain a "cold substance") [6,20,21]. Emergent processes are high-level patterns that emerge from the uniform, random, simultaneous, independent, and continuous interactions of all components in a system [20]. They result from a collection of components with equivalent status in a non-direct, disjoint manner toward a local or unintentional goal [20]. While entities (e.g., blood) and sequential processes (e.g., blood circulation) are easily observed in our everyday lives (which is why we have robust mental models of what constitutes these things), exemplars of emergent processes are not as accessible, since they often occur within temporal and spatial scales too large or small to be readily observed (e.g., evolution, molecular diffusion). So, novice learners frequently apply characteristics of direct/sequential processes to emergent processes, such as component-level interactions (that are distinct, constrained, sequential, dependent, terminating) and component-pattern relations (which exist as discrete subgroups/classes, have corresponding relationships to each other, have differential status, and a global/intentional goal) [20]. Therefore, novice learners often lack mental models of emergent phenomena, and such models must be constructed before conceptual change can take place.

Postsecondary molecular biology students frequently hold robust misconceptions about the emergent nature of molecular environments, as they have difficulty understanding how random mechanisms at the submicroscopic level can lead to macro-level, machine-like, cellular processes [2–4,9,20,25–27]. They lack a mental model of emergent processes and will apply intuitive reasoning to understand complex phenomena, either by interpreting events as goal-driven (teleological thinking) or by projecting human traits/behaviors upon molecular entities (anthropomorphic reasoning) [28,29]. The assimilation of emergent concepts with sequential processes was exemplified in a focus group discussion with undergraduate biology students [27]; participants could identify that molecules move around randomly in the cell, but they believed that this random motion is changed and becomes directed and purposeful if/when the molecules are activated by phosphorylation or receive instructions from the nucleus. Longitudinal research using the Molecular Concepts Adaptive Assessment (*n* = 1170) suggests that over 70% of first- through third-year undergraduates cannot identify random collisions as the primary mechanism of motion of a molecule toward a binding partner [27]. This finding was supported by Tobler et al. [1], who found that students believed certain molecules searched actively for interaction partners, and that active processes are necessary for binding partners to find each other. Such misconceptions often remain unaddressed throughout the students' undergraduate careers [26].

*1.2. The Role of Interactive Simulations and Game-Based Learning*

Interactive molecular simulations might support a transformative experience by serving as a visual model or exemplar of an emergent system [4,12], so that students can build a mental model of emergence, and by allowing students to confront their misconceptions head-on through manipulating the simulations and testing their hypotheses. Several researchers have suggested that such interactive learning experiences might prompt new mental model formation [4,30–34] in a process referred to as "productive negativity" (PN) [35]. Essentially, to experience PN is to learn from failure: (1) the student interacts in the environment in a way that demonstrates their misconception; (2) they realize that their way of thinking is incorrect when they are unsuccessful in a task (the negative experience); and (3) they then re-evaluate their understanding and modify their interactions to complete the task successfully (the productive response). In doing so, they begin to build a new mental model of the concept in question. The inclusion of game design might enhance the effect of PN by creating clear "fail" or negativity states, delaying advancement in the game

when understanding is not demonstrated, and using players' preconceptions about game mechanics to coax them into cycles of PN [36–38].

Importantly, a GBL intervention also has the potential to enhance the motivation and engagement needed for conceptual change to take place. A core benefit of GBL is its potential to increase motivation and engagement in learners through supporting individuals' self-determination. Self-determination theory assumes that motivation arises based on an individual's three basic social-cognitive needs: competence (mastery or perceived abilities in a task), autonomy (control or self-determining behavior), and relatedness (belonging to a group and being supported) [39,40]. The characteristics of good video games naturally fulfill these motivational requirements of self-determination theory. For example, Proulx et al. [41] describe that game mechanics such as action points, assessment, rewards, and time pressure can provide low to moderate autonomy and can be extrinsically motivating, while mechanics such as role-playing, selection/collection, competition/cooperation, progression, ownership, and strategy/planning can be highly autonomous and provide intrinsic motivation [41]. Time pressure, progression and competition can be associated with challenge in games, while others (e.g., selection, ownership, strategy, role-playing) can be associated with end-user control.

Supporting this, Wilson et al. [42] argue that challenge and motivation in video games are intertwined; an optimal level of challenge brings rise to uncertainty in goal attainment, which sustains attention until the goal is met, thus increasing players' confidence in their abilities. For example, Shute et al. [43] found that frustration positively influenced in-game performance (which in turn impacted learning outcomes), suggesting a direct link between frustration whilst engaged in challenging tasks, motivation, and achievement; the authors relate this finding back to Gee's concept of "pleasurable frustration" in video games, where game challenges are calibrated to the competency level of the player [43,44]. Similarly, Hoffman and Nadelson [45] found challenge to be a primary factor motivating prolonged video game engagement, even in the face of failure and frustration (an important consideration when implementing PN for conceptual change). Furthermore, Garris et al. [46] suggested that, in addition to challenge, clearly defined hierarchical goals in a game increase motivation and perseverance in-game by allowing for a large degree of self-determining actions or control by the player.

However, the effectiveness of GBL is thought to be heavily influenced by an individual's "gaming proficiency, personality, preferences and emotional state" [47], so a deeper investigation of how these factors are related to GBL outcomes is merited. For example, gender gaps may exist in GBL research. Males are often found to play more video games and outscore females while playing [48–50], though these difference do not always manifest in educational games, e.g., [51]. Another factor worth considering is English fluency; in 2020, 530,540 international students (66% originating from South and East Asia) were studying in Canada—a 135% increase from 2010 [52]. Furthermore, 44% of people living in Toronto have a mother-tongue other than English or French [53], and 58% of secondary schools in Ontario require English-as-a-second-language support [54]. Therefore, whether English (the language of the assessments and interventions) is a students' native language may prove to be an important predictor of outcomes. Additionally, students' academic achievement and engagement/interest in their biology course may be directly related to their misconceptions and how receptive they are to learning related content [55]. Finally, prior experience playing video games (i.e., gaming habits) may have a direct impact on students' receptiveness to a game-based learning experience [56,57]; past research by the author and colleagues found no influence of gaming habits on students' voluntary use of a serious game in one study [51] and a direct relationships between breadth of game completion and gaming habits of students in a different study [34], though neither study demonstrated a relationship between gaming habits and actual learning outcomes.

Lastly, the context of play (whether mandatory or voluntary) should be considered. Play in itself is a voluntary activity [58], which conflicts with the instructional purpose of educational games, possibly hindering engagement in the learning activity [59,60].

However, Kuindersma et al. [61] showed that mandatory play did not negatively impact post-secondary students' learning about aviation and resulted in longer total playtime when compared to students who used the game voluntarily. While greater internal validity is achieved through mandatory, monitored play, greater ecological validity is potentially achieved through experiments allowing voluntary play in out-of-class contexts. Importantly, a voluntary-play context ensures that students play willingly, which is necessary because conceptual change relies on the student having the motivation to modify their mental models [23,24,30].

### 1.3. Building on Previous Work

The GBL environment, MolWorlds, employs game mechanics in a manner that encourages failure and negativity, in hopes of instigating productive thinking about molecular emergence [33,34,62]. Pilot research found that both the GBL environment and an interactive simulation without game design (SIM; MolSandbox) resulted in significantly greater resolved misconceptions than a baseline (no-intervention) group, after 30 min of play time. A qualitative analysis of video gameplay data ($n = 42$) revealed that a significantly higher frequency of PN was experienced by GBL participants in comparison to SIM participants [33]. Furthermore, we found a trending correlation between post-intervention misconceptions and the quality of PN experienced by GBL participants ($p = 0.066$), whereas those who used SIM demonstrated no relationship between their interactions and resultant misconceptions. These findings suggest that exposing students to frustration, failure, and other negativity might be a viable means of inducing conceptual change when the experience is supported by game design. However, this previous work did not investigate potential effects that the interventions may have had on students' long-term retention, which is a better predictor for the impact of a transformative experience and was identified as a gap in the literature by the National Research Council [15] and others [14,16].

Moreover, the previous small sample did not allow us to consider how possible moderating individual or student-level attributes (e.g., gender, native language, academic achievement, course engagement, and gaming habits) might influence misconceptions or their relationship with PN [33,34]. Furthermore, though overall participation was voluntary, these prior investigations were undertaken with students who used the interventions for a mandatory 30 min exactly (high internal validity); this also led to increased boredom and disengagement in the SIM condition because some individuals repeated easy levels to fill up the time, rather than complete more difficult ones, when they would have otherwise ceased using it (low ecological validity).

### 1.4. Research Aim, Questions, and Hypotheses

The aim of the current research is to confirm or refute (with a larger sample) previous findings regarding the value-added benefit of game design for facilitating conceptual change through PN, in the context of robust misconceptions about molecular emergence. Importantly, it also targets distinct gaps in the GBL literature regarding (i) implementing a voluntary-play context, (ii) assessing GBL's impact on long-term retention of learning outcomes, and (iii) understanding what student-level factors (i.e., individual traits including gender, native English-speaking, course engagement, academic achievement, and gaming habits) might mediate GBL's effectiveness.

This was achieved by comparing learning outcomes, interaction patterns, and engagement in a GBL environment (MolWorlds)—that implements game mechanics in an attempt to facilitate PN and, thus, conceptual change—to a very similar interactive simulation environment (SIM; MolSandbox), as well as to a non-randomized baseline group from the same population (no intervention). Possible moderating attributes of the participants were controlled for, and their impacts are discussed. The study was structured to answer the following research questions (RQ):

**RQ1:** *What are the immediate and delayed value-added effects of game design on undergraduates' conceptual understanding of molecular emergence?*

It was hypothesized that GBL participants would resolve more misconceptions than those exposed to the SIM intervention, and that both interventions would be more effective than standard molecular biology education (i.e., no intervention). It was expected that retention at a delayed one-year follow-up would be higher in the GBL group in comparison to the SIM group, and that retention in both intervention groups would be higher than the baseline group.

**RQ2:** *How do interaction patterns differ between the GBL and SIM, and how are these related to conceptual change?*

It was hypothesized that the SIM group would engage in more overall experimental interactions, while the GBL group would engage in more instances of PN, due to MolWorlds' targeted implementation of gaming patterns and greater interactive freedom in MolSandbox (consistent with previous work [33]). It was expected that the higher the quality of participants' productive response to negativity (i.e., the more they demonstrate correct conceptual understandings following a PN event), the greater the conceptual change, particularly in the GBL group [34].

**RQ3:** *How does engagement in the interventions differ between the GBL and SIM groups?*

It was hypothesized that the GBL environment would generate greater engagement and motivation to play, identified by longer voluntary time-on-task and more favorable outcomes on Likert-scale assessments of motivation, due to MolWorlds' greater level of challenge, hierarchical goals, progression, and reward systems.

For all the above RQs, potential moderating individual attributes (gender, native English-speaking, course engagement, academic achievement, and gaming habits) were included in the analytical models when appropriate. Specifically, it was predicted that: (i) course engagement and academic achievement might be positively correlated with a reduction in misconceptions across all stimulus groups; (ii) there would be an interaction effect of gaming habits with stimulus, meaning that the GBL environment would be more effective and engaging for frequent game-players, but that there would be no effect of gaming habits on misconceptions in other groups; (iii) based on findings from pilot research, and contrary to reported studies suggesting that males may have more positive responses to games than females, gender would not hold an interaction effect with stimulus type in either learning or engagement outcomes; and (iv) individuals who have learned English as a first language (i.e., native-English-speakers: NES) might have overall better test scores, since non-native English-speakers (NNES) might be disadvantaged by completing English assessments and/or be less motivated to interact with English-based interventions.

## 2. Materials and Methods

Please refer to the Supplementary Materials document when section numbers are referenced and preceded by the prefix 'S'.

### 2.1. Participant Recruitment

Participants were second-year biology students at the University of Toronto Mississauga enrolled in an introductory Molecular Biology course (maximum enrolment: *n* = 480). They were recruited through a brief two-minute announcement by an investigator at the start of their lecture in the second week of term. A hyperlink to the study registration website was made available to them through the course's online portal.

### 2.2. Game and Simulation Stimuli

MolWorlds is a computer-based, 13-level, GBL environment designed to help undergraduate biology students grasp the complexity of molecular dynamics, by immersing them in a narrative-driven, adventure, simulation game that approximates a 2D-platformer genre. The core simulation allows users to modify molecular concentrations, as well as

environmental temperature and crowding, to observe their effects on rates of molecular interactions. The overarching game patterns in MolWorlds include:

- Resource management, e.g., collect molecules and power-ups (needed to modify temperature and crowding);
- Exploration/navigation;
- Sequential level progression;
- Scoring and feedback system.

These patterns are all intended to promote PN and, in turn, facilitate conceptual change. The design of MolWorlds—and specifically how learning mechanics are integrated with gaming mechanics to promote PN and engagement—has been extensively described in Gauthier & Jenkinson [33], using the activity theory model of serious games (ATMSG) framework [63]. To summarize, the game immerses the player as a third-person character in a molecular world where they are consistently confronted with the emergent nature of this environment. To complete certain molecular tasks, they will inevitably perform actions that are consistent with their misconception that molecules behave as part of a direct/sequential system. For example, to open a ligand-gated membrane channel (akin to a locked door), the player might release a single ligand (the key), expecting it to bind directly to the channel and open the way forward. A "negative" experience occurs when these actions do not result in the expected outcome, e.g., the single ligand moves off in a random direction away from the channel (this demonstrates how the resource-management gaming pattern can promote PN, by letting the player choose how many collected ligands to re-release into the environment). Now, the player must begin to reconstruct their mental model of molecular behavior and demonstrate their new understanding to proceed in the game. For instance, they could either increase the concentration of ligands, increase the temperature, or increase the crowding of the environment—or a combination of the above—to improve the probability of a ligand binding to the channel; this is the "productive" response to the negative experience.

Other game patterns are also designed to promote PN. The exploration pattern makes the player navigate through a chaotic environment that bumps and jostles them around (negative), so, they might make modifications to the emergent system to facilitate their exploration (productive). Sequential level progression encourages PN, as the player must reach a checkpoint in every level and, if not reached (negative), they may have to restart the level and reflect on what to do differently to ensure their success (productive). Lastly, scoring and feedback is presented at the end of every level by displaying 1–3 stars based on time to level completion, with an indication of how to perform better (e.g., "finish in X seconds to get 3 stars!"); upon completing a level with one star (negative), the player may be tempted to replay the level while optimizing conditions in the emergent environment (productive) to achieve 3 stars.

MolSandbox, the interactive "control" simulation environment (SIM), is also described in Gauthier & Jenkinson [33], and detailed comparisons between the ATMSG frameworks of the two stimuli are made. SIM covers the same academic content and simulated interactions as MolWorlds, but the gaming patterns listed above are removed. Removing these patterns creates inherently less challenge in the SIM environment but also creates greater freedom to experiment, since the learner faces far fewer restrictions (e.g., having to find and collect molecules and power-ups [33]). Notably, level completion in the GBL intervention requires the player to explore with the character and discover the goal of the level on their own (e.g., facilitate vesicle formation and docking), while in the SIM, the goal is written in plain language at the top of the screen. The removal of these gaming patterns, on the one hand, may decrease motivation by minimizing the user's self-determining actions in goal discovery and attainment, yet, on the other hand, may increase engagement by allowing greater control/experimentation [46].

Comparative screenshots of the interventions are shown in Figure 1; videos of gameplay for both interventions are available at sciencevis.ca/index.php/portfolio/molworlds/ (accessed on 27 March 2024).

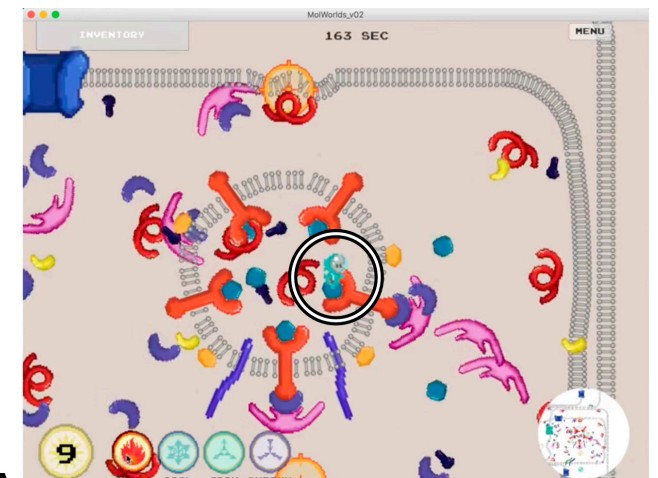 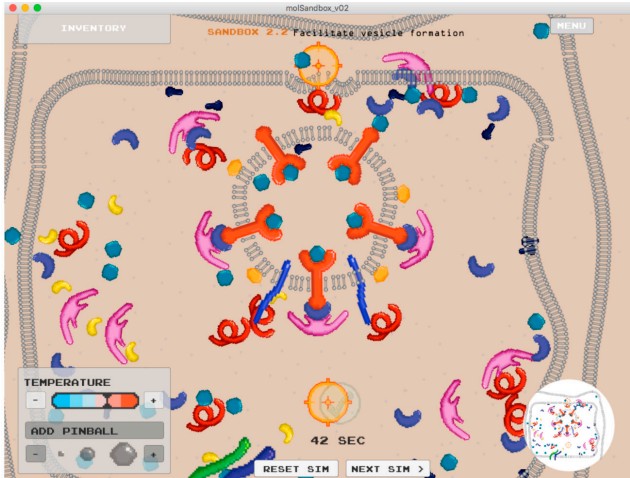

**Figure 1.** Screenshots of Level 7 in both interventions, where the goal is to facilitate cargo vesicle formation and docking. (**A**) MolWorlds screenshot, with character circled. Power-ups are displayed at the bottom-left. The player must navigate the character inside the vesicle to get to the other side of the membrane and continue the journey. (**B**) MolSandbox screenshot. Instead of power-ups, the participant uses gauges to modify the simulation. The simulation is marked as completed once the vesicle has formed and docked on the opposite membrane.

### 2.3. Pre-Intervention Assessment

**Demographics questionnaire**. The pre-test demographics questionnaire (Section S1.1.1) collected information on participants' gender identity, gaming habits, self-reported grade point average (GPA) from their previous year of study, and whether they were native English-speakers (NES) or non-native English-speakers (NNES).

**Science literacy and bioliteracy assessments**. Drawing on existing validated tools, a basic 10-question multiple-choice science literacy assessment [64] and a 10-question bioliteracy assessment [65] were implemented. The results of these assessments were only used to ensure equality in baseline knowledge between experimental groups in this publication. Their relationship with molecular misconceptions in general, removed from the context of game-based learning, is discussed by Gauthier et al. [27].

**Molecular Concepts Adaptive Assessment**. Conceptual change was measured through resolution of misconceptions from pre-test to post-test on the Molecular Concepts Adaptive Assessment (MCAA). The assessment, which is validated and described in detail in Gauthier et al. [27], asks a series of multiple-choice and true-or-false questions regarding students' perceptions of molecular behavior, e.g., "An extracellular molecule tries to move toward a complementary receptor" (True/False), and directs respondents who answer incorrectly to a second question that provides more detail about the nature of the misconception, e.g., "Based on your previous answer and assuming there are several of the complementary receptors present, an extracellular molecule tries to move toward: (a) One specific predetermined complementary receptor; (b) Any of the complementary receptors that are present; or (c) Whichever complementary receptor is closest". A maximum of 11 misconceptions is possible. Follow-up or characterizing questions, such as in the example above, are not counted in the total score. Section S1.2 displays each item included on the MCAA and how they align with the component-level interactions and component-pattern relations of emergent versus directed processes.

### 2.4. Post-Intervention Assessment

**Demographics questionnaire**. A short post-test demographics questionnaire (Section S1.1.2) asked students what grade they expected to achieve in the course, as well as to rate their engagement in their biology course on four 5-point Likert-scale questions drawn from the Burch Engagement Survey for Students [66].

**Science literacy, bioliteracy, and MCAA**. The same three assessments were implemented on the post-test following the same procedure as the pre-test.

**Intervention engagement questionnaire**. Since a voluntary-play context was used, participants were asked to select the most applicable reason for why they chose to stop using their intervention. Possible responses included: (A) I only stopped because the research study time was running out (otherwise I would have continued); (B) I have somewhere else to be, so I had to stop (otherwise I would have continued); (C) I felt that I got all I could out of it (or I finished it); (D) I had enough for one sitting (or I'm tired); (E) I didn't feel engaged enough to continue; (F) I was too frustrated with it, so wanted to stop; or (G) Some other reason. Additionally, 15 Likert-scale questions were adapted from the Instructional Materials Motivation Survey (IMMS, [67]) that could best be applied to a game and interactive simulation (Section S1.3). Five questions assessed how each of the interventions captured attention, four assessed relevance, four assessed participants' confidence in using the interventions, and three assessed participants' overall satisfaction.

### 2.5. Delayed Follow-Up Assessment

This assessment consisted of the same science literacy, bioliteracy, and MCAA components as the post-intervention assessment.

### 2.6. Protocol

#### 2.6.1. Survey-Only Protocol (Baseline Group)

Second-year biology students completed the pre-test assessment starting in the second week of term on a password-protected study website. Students used their university credentials to access the site and all responses were stored in a MySQL database. Registration remained open for two weeks. For those who completed all aspects of the pre-test, a link to the post-test assessment was emailed in the second-to-last week of term, before the exam period. The post-test also remained open online for two weeks. Students who completed the post-test were rewarded with a 0.5% bonus mark in their course. Completion of all aspects of the surveys took approximately 10–15 min each at both pre-test and post-test.

#### 2.6.2. Randomized Controlled Trial Protocol

Those who completed all aspects of the pre-test were sent—by email one month before the end of term (i.e., in advance of the post-test)—an invitation to participate in the randomized controlled trial (RCT). Participation in the RCT was compensated with a $10 gift card to an on-campus coffee shop (in addition to the 0.5% bonus mark) and students were told that the study would run for approximately 60 min (though the actual time for each participant varied, as described below). Students registered for the RCT through an online form for one of several timeslots over a two-week period toward the end of term; RCT-registered students were unable to complete the post-test prior to participating in the RCT.

The RCT took place in an on-campus computer lab. Participants were notified that they would play with one of two computer applications that applied game design to various extents; in this way, it was hoped to blind those assigned to the SIM condition from knowing that they did not receive the "real game" version. Upon logging into the pre-loaded study website, participants were computationally randomized into either Condition 1 (SIM) or Condition 2 (GBL) and the allocation (Condition 1 or 2) was displayed at the top of the screen. The randomization process was minimized for gender to achieve equal distribution between groups [68]. Participants first completed the brief post-test demographics questionnaire (~2 min), after which a study moderator opened their assigned intervention and began recording their screens using QuickTime Player [69]. The protocol strives to achieve a balance between ecological and internal validity by requiring a minimum playtime (20 min), thereby ensuring that enough data are collected in a controlled environment but allowing the participants to choose to continue playing afterward, to approximate ecologically valid conditions. As such, participants were instructed that they could use their intervention

between a minimum of 20 min and a maximum of 45 min, to complete the study within a 60 min time-frame; before beginning, they were instructed to play as they would if they were at home, to not feel pressured to finish all the content, and to stop any time after 20 min. The time at which they could cease play (current time + 20 min) was written by a study moderator on a sheet of paper in front of the participant once the screen recording was started; students had access to a clock in the room.

During the RCT, students' interactions in the intervention (e.g., collecting or releasing molecules, increasing temperature, etc.; see Section S1.4 for a list of data) were collected into the MySQL database. Participants notified the moderator when they wanted to stop by raising their hand; the screen-recordings were saved, and the participant then completed all other aspects of the post-test survey in the same fashion as the baseline group. However, RCT participants additionally completed the intervention engagement survey, which baseline participants did not.

### 2.6.3. Delayed Follow-Up Protocol

One year following the pre-test, all participants (from both RCT and baseline groups) who completed both the pre- and post-tests in the previous year were invited by email to complete the assessment one last time. They were incentivized with a chance to win 1 of 20 $10 gift cards to an on-campus coffee shop.

## 3. Data Analysis and Results

### 3.1. General Approach

All analyses were performed in SPSS Statistics v.24.0.0 [70], using target $\alpha = 0.05$. Science literacy, bioliteracy, and MCAA were automatically graded with a PHP script that drew answers directly from the database, to remove the possibility for marker error. For continuous data that did not follow a normal distribution, the Shapiro–Wilk test results are presented, and non-parametric investigations were performed in lieu of parametric tests. The student attribute variables of "course engagement", "academic achievement", and "gaming habits" were derived by reducing multiple dimensions from the demographic questionnaires (Section S2).

### 3.2. Participant Composition

A total of 278 second-year biology students completed the pre-test and 224 also completed the post-test (19.43% pre-post attrition, 46.66% final class participation). Those who did not complete the post-test were excluded from the analyses in this paper, as were another three students who had previously taken part in the pilot research [34]. A portion of these same students ($n = 84$), participated in the RCT. This resulted in 42 GBL participants, 42 SIM participants, and 138 non-randomized baseline participants in total. A MANOVA and multinomial logistic regression were performed to assess the homogeneity of group composition prior to intervention exposure. Results revealed no differences in science literacy, bioliteracy, pre-test misconceptions, course engagement, academic achievement, or gaming habits between groups, as well as an even distribution of native- and non-native English-speakers, and males and females, across groups (Section S3).

At the one-year delayed follow-up time point, only 71 participants were retained (68.3% overall attrition): 36 baseline participants (73.9% loss), 18 SIM participants (57.1% loss), and 17 GBL participants (59.5% loss). While the amount of attrition at the delayed follow-up is considered to be significant [71], tests for the effect of attrition on misconception scores (Section S4.2) would suggest that attrition bias is minimal.

### 3.3. RQ1: What Are the Immediate and Delayed Value-Added Effects of Game Design on Students' Conceptual Understanding of Molecular Emergence?

#### 3.3.1. Immediate Effect on Conceptual Change

Unlike conventional test scores, misconceptions were the unit of measure on the MCAA; therefore, lower scores represent more favorable outcomes. A repeated measures

ANCOVA was performed to investigate the effect of MolWorlds and MolSandbox on students' misconceptions in comparison to the baseline group, whilst controlling for gender, native language and, as covariates, course engagement, academic achievement, and gaming habits. In addition to the main effects of all variables, full-factorial interaction effects between categorical variables were included in the model, plus the interaction between the gaming habits covariate and stimulus group. Pairwise comparisons with Sidak adjustments were planned for any significant main effects and/or interaction effects. Table 1 summarizes the scores achieved on the MCAA by stimulus group and by language-stimulus sub-groups.

**Table 1.** Unadjusted misconception scores measured by the MCAA.

| | | PRE-TEST | | | POST-TEST | | | CHANGE | | |
|---|---|---|---|---|---|---|---|---|---|---|
| **Group** | **N** | **Mean** | **SD** | **Min, Max** | **Mean** | **SD** | **Min, Max** | **Mean** | **SD** | **Min, Max** |
| **Baseline overall** | **138** | **5.59** | **2.26** | **0, 10** | **5.34** | **2.28** | **0, 11** | **−0.25** | **2.57** | **−6, 5** |
| Baseline NES | 87 | 5.36 | 2.26 | 0, 9 | 5.30 | 2.15 | 1, 9 | −0.06 | 5.59 | −6, 5 |
| Baseline NNES | 51 | 6.00 | 2.22 | 1, 10 | 5.41 | 2.49 | 0, 11 | −0.59 | 2.52 | −6, 5 |
| **SIM overall** | **42** | **5.67** | **2.39** | **0, 10** | **4.50** | **2.82** | **1, 10** | **−1.17** | **3.00** | **−7, 4** |
| SIM NES | 24 | 5.54 | 2.67 | 0, 10 | 5.21 | 3.05 | 1, 10 | −0.33 | 2.90 | −7, 4 |
| SIM NNES | 18 | 5.83 | 2.01 | 1, 8 | 3.56 | 2.23 | 1, 8 | −2.28 | 2.85 | −7, 2 |
| **GBL overall** | **42** | **6.12** | **2.34** | **1, 10** | **4.36** | **2.85** | **0, 9** | **−1.76** | **2.55** | **−7, 2** |
| GBL NES | 29 | 6.38 | 2.21 | 2, 10 | 4.03 | 2.82 | 0, 9 | −2.35 | 2.54 | −7, 2 |
| GBL NNES | 13 | 5.54 | 2.60 | 1, 9 | 5.08 | 2.90 | 1, 9 | −0.46 | 2.11 | −5, 2 |

NES = native-English speakers; NNES = non-native English-speakers.

The results of the repeated measures ANCOVA are reported in Table 2 and visualized in Figure 2. There was an overall reduction in misconceptions over the course of the semester. A significant effect of the stimulus group was observed on conceptual change, as was the interaction between stimulus group and native language. In other words, stimulus groups improved at differential rates and the effect was not consistent across native- (NES) and non-native English-speakers (NNES). There were no main effects of gender or native language on conceptual change, nor were there interaction effects between stimulus group and gender or between stimulus group, gender, and native language. Furthermore, there were no significant moderating effects of course engagement (trending), gaming habits, or academic achievement on conceptual change. Lastly, gaming habits did not have an interaction effect with stimulus type, contrary to the authors' predictions.

**Table 2.** Repeated measured ANCOVA: the effects of stimulus intervention on change in misconceptions, whilst controlling for potential moderators.

| | **df** | **F** | **p** | **Partial $\eta^2$** | **Obs. Power** |
|---|---|---|---|---|---|
| Change | 1 | 18.79 | <0.001 * | 0.084 | 0.991 |
| Change × course engagement | 1 | 3.50 | 0.063 | 0.017 | 0.461 |
| Change × gaming habits | 1 | 0.02 | 0.886 | 0.000 | 0.052 |
| Change × academic achievement | 1 | 2.27 | 0.133 | 0.011 | 0.323 |
| Change × stimulus | 2 | 4.78 | 0.009 * | 0.045 | 0.790 |
| Change × gender | 1 | 0.73 | 0.393 | 0.004 | 0.136 |
| Change × native language | 1 | 0.54 | 0.463 | 0.003 | 0.113 |
| Change × stimulus × gender | 2 | 1.25 | 0.289 | 0.012 | 0.270 |
| Change × stimulus × native language | 2 | 3.27 | 0.040 * | 0.031 | 0.616 |
| Change × gender × native language | 1 | 1.03 | 0.311 | 0.005 | 0.173 |
| Change × stimulus × gender × native language | 2 | 1.20 | 0.304 | 0.012 | 0.260 |
| Change × stimulus × gaming habits | 2 | 1.30 | 0.275 | 0.013 | 0.279 |

Type III sum of squares was used. Significant factors are highlighted (*); Box's test: $M = 28.00$, $F(33, 3771.73) = 0.76$, $p = 0.839$; Levene's pre-test: $F(11, 209) = 0.87$, $p = 0.568$; Levene's post-test: $F(11, 209) = 1.07$, $p = 0.384$.

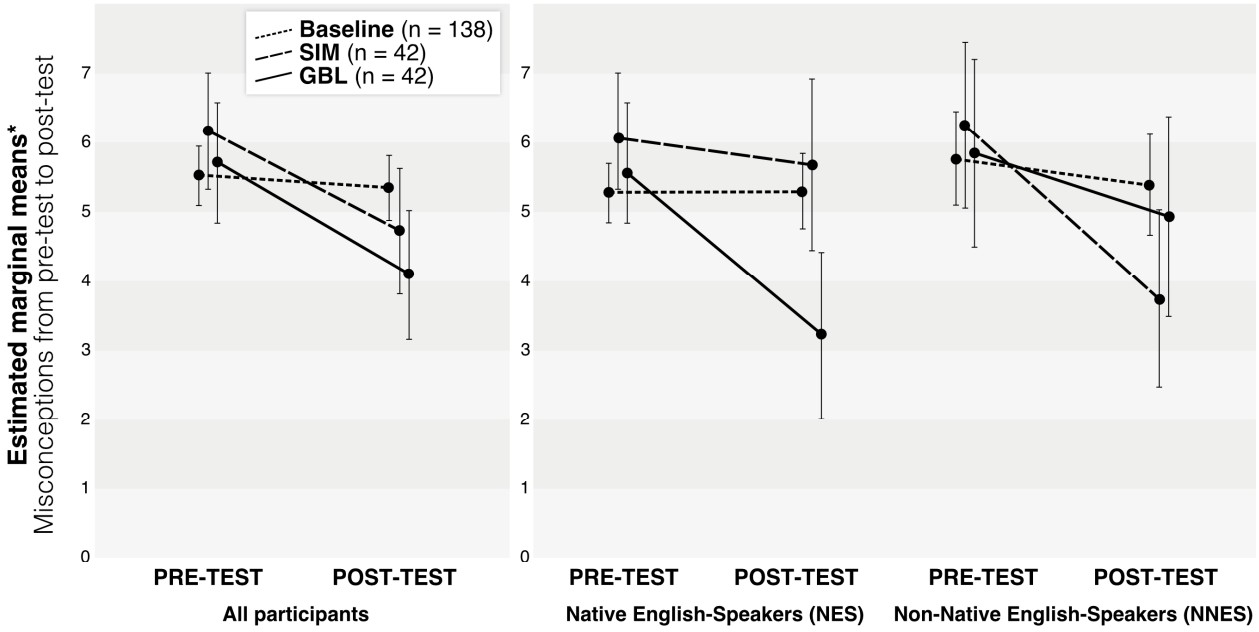

*Covariates appearing in the model are evaluated at the following values:
Course engagement = -0.0024; Gaming habits = 0.0056; Student achievement = -0.0076

**Figure 2.** Estimated marginal mean misconceptions at pre-test and post-test time points, controlling for potential moderators. Error bars = 95% confidence intervals.

Sidak-adjusted pairwise comparisons evaluated covariates at course engagement of −0.002, gaming habits of 0.006, and academic achievement of −0.008. These analyses revealed that, at the stimulus level, SIM participants improved only a trending amount more than baseline students ($M_{diff}$ = −1.27, 95% CI [−2.61, 0.07]; $p$ = 0.069), while GBL participants improved significantly more than baseline ($M_{diff}$ = −1.44, 95% CI [−2.81, −0.08]; $p$ = 0.035). No difference was found between SIM and GBL conditions ($M_{diff}$ = −0.18, 95% CI [−1.87, 1.52]; $p$ = 0.992). However, further Sidak-adjusted, post hoc analyses investigating the interaction between stimulus and native language reveal a more nuanced understanding of these changes. SIM-NNES participants ($M_{diff}$ = −2.52, 95% CI [−4.76, −0.28]; $p$ = 0.015) and GBL-NES participants ($M_{diff}$ = −2.36, 95% CI [−4.46, −0.25]; $p$ = 0.016) resolved significantly more misconceptions than baseline-NES participants. No other comparisons were significant (Section S4.1).

### 3.3.2. Delayed Effect on Conceptual Change

One year after the pre-intervention assessment was distributed, baseline participants held 4.89 (SD = 2.76) misconceptions, while SIM participants held 4.56 (SD = 2.18) misconceptions and game participants held 3.71 (SD = 2.37) misconceptions. Due to the small size of the retained sample at the delayed follow-up time point, there were only enough degrees of freedom to include a single variable (stimulus group) in a repeated measures ANOVA. Additionally, though the distribution of NNES participants was statistically equal between groups ($\chi^2(2)$ = 2.90, $p$ = 0.235), only three participants in the GBL group came from this minority at the delayed follow-up, which makes further analyses into stimulus-language subgroups inappropriate.

The model showed an overall significant reduction in misconceptions over time ($F(2)$ = 9.95, $p < 0.001$, $\eta^2_p$ = 0.128) and a significant interaction effect with stimulus group ($F(4)$ = 2.74, $p$ = 0.031, $\eta^2_p$ = 0.075), meaning that differences in conceptual change-retention existed between stimulus groups. The results are visualized in Figure 3. Within subjects' contrasts (Section S4.2) show a trending linear effect for the interaction between change over time and stimulus group (i.e., the difference between pre-test and delayed follow-up-test scores; $F(2)$ = 2.54, $p$ = 0.086). Even though this result is not statistically significant, it was

believed that it warranted closer investigation since it was achieved with a small sample. Sidak-adjusted pairwise comparisons suggest that the primary difference lies between the GBL group and baseline group ($M_{diff}$ = −1.62, 95% CI [−3.38, 0.14]; $p$ = 0.080), rather than between SIM and baseline groups ($M_{diff}$ = −0.50, 95% CI [−2.23, 1.23]; $p$ = 0.860) or between GBL and SIM groups ($M_{diff}$ = −1.12, 95% CI [−3.14, 0.90]; $p$ = 0.448).

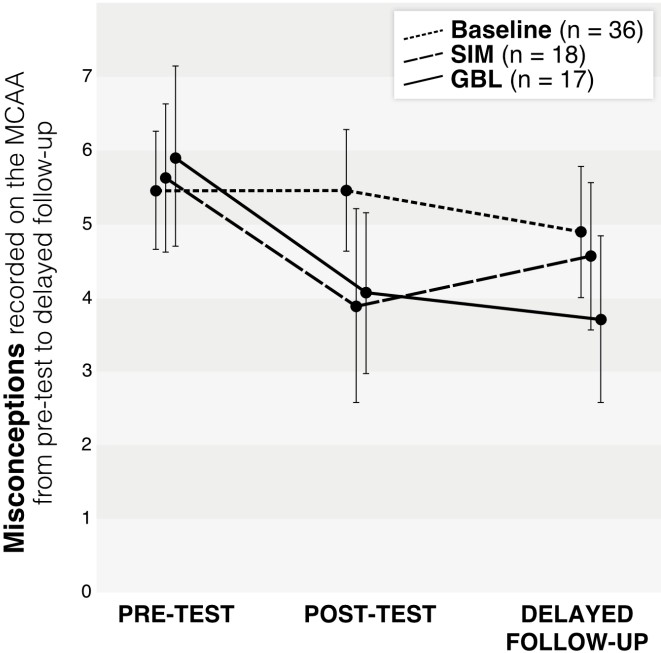

**Figure 3.** Estimated marginal mean misconceptions at pre-test, post-test, and delayed one-year follow-up time points. Error bars = 95% confidence intervals.

### 3.4. RQ2: How Do Interaction Patterns Differ between the GBL and SIM Interventions, and How Are These Related to Conceptual Change?

Gameplay in the RCT intervention groups was analyzed through two methods: (1) a quantitative analysis investigating differences in "click-stream" interaction data, and (2) a qualitative analysis of gameplay behavior by coding video screencasts with demonstrations of correct conceptual knowledge and instances of PN. From this qualitative analysis, the "PN quality" metric was computed, which was hypothesized would be related to conceptual change.

#### 3.4.1. Quantitative Interaction Data

Summative interaction data failed Shapiro–Wilks' test of normality (all $p$ < 0.05). Consequently, non-parametric Mann–Whitney U tests were performed.

The interaction data show that MolSandbox allowed for more interactive freedom than MolWorlds, as evidenced by greater average levels attempted, levels completed, unique levels completed (breadth of intervention completion), molecules released, temperature modifications, and molecule information accessed (Table 3). Bivariate Spearman correlations were performed between each interaction statistic and gaming habits (Section S6). The gaming habits of SIM participants were significantly correlated with the number of successful level completions ($r_{Sp}$ = 0.44, $p$ = 0.004) and the number of unique levels completed ($r_{Sp}$ = 0.41, $p$ = 0.006). In the GBL group, positive correlations existed between gaming habits and most interaction statistics, including total levels completed ($r_{Sp}$ = 0.38, $p$ = 0.012), power-ups (temperature and crowding modifications) used ($r_{Sp}$ = 0.42, $p$ = 0.006), and molecule information accessed ($r_{Sp}$ = 0.35, $p$ = 0.024). Notably, the relationship between unique levels completed (breadth of game completion) was only tentatively associated with gaming habits in the MolWorlds group ($r_{Sp}$ = 0.27, $p$ = 0.086), unlike pilot research that demonstrated a significant relationship.

**Table 3.** Telemetry data: average frequency of various interactions performed in MolWorlds and MolSandbox.

| Interaction | MolWorlds (GBL) | | | MolSandbox (SIM) | | | Difference | | | |
|---|---|---|---|---|---|---|---|---|---|---|
| | Mean | SD | Min, Max | Mean | SD | Min, Max | U | Z | p | + |
| Levels started | 11.07 | 3.21 | 6, 23 | 25.19 | 11.83 | 10, 66 | 97.50 | −7.03 | <0.001 | S |
| Levels completed | 8.07 | 2.11 | 5, 15 | 13.95 | 4.15 | 7, 28 | 117.50 | −6.87 | <0.001 | S |
| Unique levels completed | 7.24 | 1.38 | 5, 10 | 10.88 | 1.77 | 6, 13 | 120.50 | −6.89 | <0.001 | S |
| Collect molecules | 115.02 | 69.45 | 8, 343 | 136.81 | 96.46 | 10, 448 | 789.50 | −0.83 | 0.408 | - |
| Release molecules | 19.64 | 9.40 | 4, 47 | 32.14 | 11.56 | 7, 59 | 333.50 | −4.91 | <0.001 | S |
| Temperature mods. | 18.05 | 17.19 | 0, 71 | 83.07 | 46.30 | 6, 183 | 129.00 | −6.74 | <0.001 | S |
| Crowding mods. | 5.91 | 3.62 | 2, 19 | 4.52 | 3.23 | 0, 11 | 729.50 | −1.37 | 0.170 | - |
| Item info accessed | 22.36 | 19.76 | 1, 104 | 78.33 | 42.29 | 20, 244 | 128.50 | −6.74 | <0.001 | S |

+ indicates whether greater interactions are seen in the simulation (S) or GBL (G) group for significant ($p < 0.050$) results, for ease of viewing.

3.4.2. Qualitative Gameplay Data

NVIVO for Mac v.11.4.1 [72] was used to qualitatively analyze video-footage of gameplay, using a deductive approach to thematic analysis [73], as employed and described in the author's previous research [33]. There was a problem with one video in the GBL group (it was cut short at 23 min, but their gameplay was telemetrically recorded for 31.60 min in the database), so this individual was excluded from analyses involving qualitative data. All interactions made in the videos (refer to Section S1.4) were transcribed. Transcribed interactions were then coded as either a demonstration of correct conceptual knowledge (DCCK) or an instance of productive negativity (PN) (defined below) by a primary coder, while interactions that were not productive or did not constitute a negative event were left un-coded. A secondary rater coded 25% of the videos, whilst blinded to the assessment of the primary rater, to ensure the accuracy of the primary rater's assessment [74], using an intra-class correlation coefficient. Detailed results of interactions that constitute DCCK/PN categories are provided in the coding scheme in Section S7 and results in Section S8.

**Demonstrations of correct conceptual knowledge**. DCCK are defined as adjustments made in the SIM/GBL environments that directly benefited goal completion [33,34]. These included changes in temperature (chill or heat environment), crowding (increase/decrease pinball/character size), and concentration (collect or release molecules) in specific contexts. For example, if the level required one cargo molecule to bind to one receptor, then (a) increasing the concentration of cargo molecules by releasing several from the inventory and (b) increasing the temperature to facilitate faster interactions, would constitute two DCCK. It is important to note that a distinction could not be made between an intentional interaction that the student knew would benefit them and a "lucky guess" or experimentation with a particular variable; both were coded as a DCCK if the action benefited goal completion, thereby indicating correct conceptual knowledge about emergent behavior. A percentage of total modifications made that were DCCKs was also calculated. For example, an individual may have modified the temperature a total of 20 times, but these actions may have only benefited goal completion 10 times (i.e., 10 DCCKs); this results in a 50% rate of total beneficial modifications in temperature. These statistics were computed to account for the greater interactive freedom of MolSandbox [33]. Comparisons across stimulus groups were made using non-parametric Mann–Whitney U tests.

Interrater reliability for DCCK was excellent (Cronbach's α = 0.967, ICC = 0.936, 95% CI [0.853, 0.973]). GBL participants averaged 19.24 (SD = 12.69) DCCKs, whereas SIM participants averaged 25.17 (SD = 7.18) DCCKs, which is a significant difference in favor of SIM participants ($U = 532.50$, $Z = −3.00$, $p = 0.003$), consistent with the higher rates of tracked interactions reported in Section 3.4.1. However, when considering the percentage of total modifications made that were productive, we observe 11.38% (SD = 3.68) productive modifications in the GBL group and 11.19% (SD = 4.45) productive modifications in the SIM group, which are similar proportions ($U = 804.50$, $Z = −0.52$, $p = 0.607$). Gaming habits were not correlated with the percentage of productive modifications in either the SIM group

($r_{Sp}$ = 0.03, $p$ = 0.852) or GBL group ($r_{Sp}$ = 0.21, $p$ = 0.193). However, gaming habits were correlated with total DCCKs in the GBL group ($r_{Sp}$ = 0.42, $p$ = 0.007) but not in the SIM group ($r_{Sp}$ = 0.02, $p$ = 0.879).

**Instances of productive negativity**. An instance of PN was identified when the participant experienced a delay in progress, level failure, or some other form of negativity, which was then compensated for by an action that benefited goal completion (i.e., one or more DCCK). Five types of negativity sources were defined through previous research: (1) resource retentiveness, (2) difficult resource collection, (3) resources lost due to overheating, (4) navigation and reaching checkpoints (MolWorlds only), and (5) simulation-driven sources of negativity [33]. An additional two categories were identified in this study: (6) scoring less than 3 stars (which was previously hypothesized to occur but did not previously observe), and (7) incorrect actions (which was not previously considered at all). Comparisons across stimulus groups were made using Mann–Whitney U tests.

Inter-rater reliability assessment on instances of PN was strong (Cronbach's α = 0.927, ICC = 0.865, 95% CI [0.703, 0.942]). Overall, there was significantly higher PN in the GBL group ($U$ = 638.50, $Z$ = −2.06, $p$ = 0.040), who averaged 4.59 (SD = 3.01) PN events, in comparison to the 3.29 (SD = 1.83) PN events experienced by the SIM group, a finding consistent with the pilot research. Gaming habits were correlated with PN in the GBL group ($r_{Sp}$ = 0.34, $p$ = 0.029) but not in the SIM group ($r_{Sp}$ = 0.21, $p$ = 0.187).

**Productive negativity quality**. The quality (or impact) of students' PN experiences was computed by dividing their total number of DCCK by their total number of PN events:

$$\text{PN quality} = \text{Total DCCK} \div \text{Total PN events}$$

This statistic was calculated to test if it was related to a reduction in students' misconceptions. The reasoning is that, if a PN event was successful at transforming misconceptions (i.e., it was impactful and, thus, of 'high quality'), we would expect several DCCK to follow the negative event, in addition to the initial productive response. Differences between SIM and GBL groups were assessed with a Mann–Whitney U test.

SIM participants scored a mean PN quality of 10.12 (SD = 6.65) and GBL participants scored a mean 4.34 (SD = 1.52), a significant difference, $U$ = 196.00, $Z$ = −6.06, $p$ < 0.001. Gaming habits were not correlated with PN quality in either the GBL group ($r_{Sp}$ = 0.23, $p$ = 0.150) or the SIM group ($r_{Sp}$ = −0.23, $p$ = 0.139). However, it is the relationship between PN quality and conceptual change that is of real interest, the results of which immediately follow in the next section.

### 3.4.3. Relationship between PN Quality and Conceptual Change

**Immediate relationship**. Preliminary bivariate correlations showed a significant relationship between PN quality and loss of misconceptions in the GBL group ($r_{Sp}$ = −0.44, $p$ = 0.004) but no relationship in the SIM group ($r_{Sp}$ = 0.13, $p$ = 0.432). Based on these results, two multivariable linear regressions were performed (one for the GBL group and one for the SIM group) to test the predictive nature of PN quality (independent variable) on conceptual change (dependent variable). To improve the interpretability of the results, the predictor variables were mean-centered before including it in the model. In these analyses, the potential interaction effect between native English-speaking and PN quality was also tested, by using the product between the dummy variable and mean-centered PN quality.

Overall, the SIM group's model was not significant ($R^2$ = 0.03; F(2, 39) = 0.16, $p$ = 0.599), with only 3% of the variance in conceptual change outcomes explained, whilst the GBL group's was significant ($R^2$ = 0.22; F(2, 38) = 5.39, $p$ = 0.009), with 22% of variance explained. Table 4 summarizes the beta coefficients for each model. For GBL participants, every unit increase in PN quality (i.e., every additional DCCK per PN experience) was related with resolving 0.74 (95% CI [0.24, 1.23]) misconceptions. So, for example, a GBL participant who experienced a total of three PN events and responded with a total of 21 DCCK (PN quality = 7.00) would be expected to resolve approximately 3.03 (95% CI [1.47, 4.59]) misconceptions, whereas a SIM participant with the same PN quality might resolve 1.39

(95% CI [0.34, 2.44]) misconceptions. The interaction effect of native language and PN quality was not significant.

**Table 4.** Multivariable linear regression analyses predicting conceptual change—coefficients.

| Stimulus | | Unstandardized Coefficients | | | | 95% Confidence Interval for B | |
| --- | --- | --- | --- | --- | --- | --- | --- |
| | | B | Std. Error | $t$ | $p$ | Lower | Upper |
| SIM | (Constant) | −1.377 | 0.512 | −2.687 | 0.011 | −2.413 | −0.340 |
| | mean-centered PN quality | 0.093 | 0.122 | 0.759 | 0.453 | −0.154 | 0.339 |
| | native language × mean-centered PN quality | −0.031 | 0.141 | −0.218 | 0.828 | −0.317 | 0.255 |
| GBL | (Constant) | −3.116 | 0.837 | −3.722 | 0.001 | −4.811 | −1.421 |
| | mean-centered PN quality | −0.735 | 0.244 | −3.013 | 0.005 | −1.229 | −0.241 |
| | native language × mean-centered PN quality | 0.425 | 0.233 | 1.823 | 0.076 | −0.047 | 0.896 |

**Delayed effect**. Finally, since the data for long-term retention were non-parametric, bivariate Spearman correlations were performed between PN quality and retention scores (differential from pre-intervention to delayed follow-up, as well as the differential from post-intervention to delayed follow-up) to understand whether similar relationships existed between PN quality, stimulus type, and conceptual change in the long-term. These did not reveal a relationship in the GBL (pre-delayed: $r_{Sp} = -0.05$, $p = 0.845$; post-delayed: $r_{Sp} = 0.30$, $p = 0.241$) or SIM (pre-delayed: $r_{Sp} = 0.22$, $p = 0.387$; post-delayed: $r_{Sp} = -0.14$, $p = 0.588$) group.

### 3.5. RQ3: How Does Engagement in the Interventions Differ?

Engagement was assessed based on (1) voluntary time-on-task and on (2) self-reported metrics of motivation measured on the post-intervention intervention engagement questionnaire.

### 3.5.1. Voluntary Time-on-Task

Voluntary time-on-task was computed by extracting the timestamp of the first logged event performed by a participant (i.e., intervention login) and subtracting this from the timestamp of the last logged event by the same participant. A univariate ANCOVA was used to investigate the relationship between time-on-task, stimulus group, and gaming habits, plus any significant variables found to affect conceptual change in previous analyses.

Participants in the GBL group averaged 30.90 (SD = 6.63) minutes of voluntary intervention usage, whereas those in the SIM group averaged 26.75 (SD = 5.51) minutes. Only the binary native-language variable (NNES/NES) was added to the model alongside stimulus group and gaming habits, since it showed significant interaction effects in previous analyses, and it is optimal to retain 10–15 observations per predictor variable [75]. The model was statistically significant ($F(7, 76) = 2.79$, $p = 0.012$, $R^2_{adj} = 0.13$, partial $\eta^2 = 0.20$, observed power = 0.89). Stimulus type was a statistically significant factor ($F(1, 76) = 8.21$, $p = 0.005$), meaning that GBL participants used their intervention longer than SIM participants. Native English-speaking did not have a main effect nor an interaction effect with stimulus (Section S9). Furthermore, while gaming habits were not an overall predictor of length of use, the interaction between gaming habits and stimulus type was ($F(1, 76) = 4.86$, $p = 0.031$). Post hoc bivariate Pearson correlations ($\alpha = 0.025$) showed a significant and moderately strong relationship between gaming habits and time-on-task in GBL participants ($r = 0.38$, $p = 0.013$) but not in SIM participants ($r = -0.09$, $p = 0.552$). This means that "gamers" voluntarily used MolWorlds for longer than non-gamers, while gaming habits did not influence use of MolSandbox.

### 3.5.2. Intervention Engagement Questionnaire

**Reasons for ceasing voluntary use**. A 2 × 5 Pearson Chi-square analysis was performed to examine the relationship between stimulus and reasons for intervention cessation, to better understand voluntary time-on-task. Reasons D (I had enough for one sitting) and

E (I didn't feel engaged enough to continue) failed to meet the criteria for a Chi-square by having an expected cell count less than 5, so, since the two reasons also reflected similar sentiments, these categories were combined for the analysis. Reason G (some other reason) was not chosen by any participant and was excluded. Post hoc analyses used the multiple regression for contingency tables approach described by Beasley and Schumacker [76]. Table 5 summarizes the distribution of participants' reasons for stopping use of their assigned intervention, as well as the results from post hoc analyses. A Sidak-adjusted $\alpha$-value of 0.0051 (10 comparisons) was used to assess significance. The overall Pearson Chi-square was significant ($\chi^2(4) = 28.04$, $p < 0.001$). Post hoc comparisons reveal that the reason "I felt that I got all I could out of it (or I finished it)" was a more frequent reason for SIM participants than GBL participants, whereas the reason "I was too frustrated with it, so wanted to stop" was a more likely choice in GBL than SIM participants.

**Table 5.** Cross-tabulation of reasons for voluntary intervention cessation in GBL ($n = 42$) and SIM ($n = 42$) conditions.

| Reason to Stop | *MolWorlds* (GBL) | | | *MolSandbox* (SIM) | | | Post Hoc | |
|---|---|---|---|---|---|---|---|---|
| | N | % | Adj. Res. | N | % | Adj. Res. | $\chi^2$ | *p* |
| (A) I only stopped because the research study time was running out (otherwise would continue) | 10 | 23.81 | 2.11 | 3 | 7.14 | −2.11 | 4.46 | 0.035 |
| (B) I have somewhere else to be, so I had to stop (otherwise would continue) | 10 | 23.81 | 0.00 | 10 | 23.81 | 0.00 | 0.00 | 1.000 |
| (C) I felt that I got all I could out of it (or I finished it) | 1 | 2.38 | −4.61 | 19 | 45.24 | 4.61 | 21.26 | <0.001 * |
| (D,E) I had enough for one sitting (or I'm tired)/I didn't feel engaged enough to continue | 8 | 19.05 | 0.00 | 8 | 19.05 | 0.00 | 0.00 | 1.000 |
| (F) I was too frustrated with it, so wanted to stop | 13 | 30.95 | 3.13 | 2 | 4.76 | −3.13 | 9.82 | 0.002 * |

The original cross-tabulation was significant ($\chi^2(4) = 28.04$, $p < 0.001$). Sidak-adjusted *p*-value of 0.0051 was used to assess significance (*) of post hoc comparisons. Adj. res = adjusted residual.

**Ordinal-scale motivations questionnaire**. Lastly, Mann–Whitney U tests were performed on each ordinal engagement IMMS statement to determine differences in agreement between stimulus groups. Medians and interquartile ranges (IQR) for each statement, as well as test result details, are given in Section S9.1. Three of the IMMS engagement statements proved significantly different between stimulus groups. Firstly, SIM participants were more likely to agree to the statement "the material covered in the intervention was more difficult to understand than I would like for it to be" (Median = 3.00, IQR = 2) than GBL participants (Median = 2.00, IQR = 1), $U = 576.00$, $p = 0.004$. Secondly, GBL participants were more likely to agree (Median = 3.00, IQR = 1) that the "exercises in the intervention were too difficult" than SIM participants (Median = 2.00, IQR = 1), $U = 639.00$, $p = 0.023$. Finally, SIM participants were more likely to agree (Median = 3.00, IQR = 2) that they "could not really understand quite a bit of the material in the intervention" than GBL participants (Median = 2.00, IQR = 1), $U = 538.00$, $p = 0.001$.

A series of additional, unplanned analyses were performed based on the finding that the interventions had opposite effects based on whether English was the native language of participants (NES) or not (NNES). Kruskal–Wallis tests on the IMMS statements revealed a difference in how stimulus-language subgroups perceived the difficulty of the material in the interventions ($\chi^2(3) = 10.71$, $p = 0.013$), whether the intervention was too abstract ($\chi^2(3) = 14.53$, $p = 0.002$), and whether they could understand the material in the intervention ($\chi^2(3) = 11.36$, $p = 0.010$). Post hoc comparisons with Sidak-corrected $\alpha$-values of 0.0085 (6 comparisons) were performed (Section S9.2). SIM-NES participants were more likely than GBL-NES participants to agree that the material in the intervention was too difficult ($U = 184.50$, $p = 0.002$), too abstract ($U = 210.50$, $p = 0.008$), and that they could not really understand what was going on in the intervention ($U = 183.00$, $p = 0.002$). Additionally, SIM-NES participants were more likely than SIM-NNES participants to perceive the intervention as being too abstract ($U = 91.00$, $p = 0.001$).

## 4. Discussion

### 4.1. GBL Increased the Effectiveness of the Simulation on Conceptual Change

Overall, participants exposed to MolWorlds resolved more misconceptions than baseline participants after a voluntary 20–45 min of gameplay. While those who used MolSandbox resolved a similar number of misconceptions in comparison to their GBL peers, their differences from those who received standard instruction (baseline group) did not quite reach statistical significance, when controlling for covariates. These findings support the author and colleagues' previous work [33,34], as well as other research touting interactive simulations, and GBL in particular, as facilitators of conceptual change [4,12,36–38]. Additionally, upon a one-year follow-up, trends suggested that those who used MolWorlds retained their conceptual change, while those who used MolSandbox did not, lending further evidence to GBL's value-added benefit. Nonetheless, not all misconceptions were resolved amongst any participant. This could perhaps be because participants did not form a robust enough conceptual model of emergent systems. Future research should consider integrating more direct teaching of what characterizes an "emergent schema" (e.g., as in [4]), rather than relying solely on students' interactions in the simulation to build up the schema from scratch. Alternatively, longer exposure to the intervention may have enabled the resolution of more misconceptions [77].

### 4.2. Moderating Effect of Native Language—A Proxy for Culture?

It was hypothesized that non-native English speakers might be disadvantaged by completing English assessments and/or be less motivated to interact with English-based interventions, and, thus, this variable was controlled for. However, it was not anticipated that the effect would be different based on the stimulus received.

Native and non-native English-speakers resolved a similar number of misconceptions overall, but there was an unexpected interaction effect of native English-speaking and stimulus group. Upon inspection, it appeared that MolWorlds was highly effective for native English-speakers, while MolSandbox was effective for non-native English-speakers. This was unexpected since the two interventions were text-light and, when text was present (e.g., molecule info panels), the interventions used the same language (the exception to this is the simple dialogue in the game's introductory narrative, and each level's goal statement in the SIM intervention). If anything, MolSandbox might have disadvantaged non-native speakers since the goal in each level is written at the top of the screen and required explicit understanding; for example, in level 7, the SIM intervention states that the goal of the level is to "facilitate vesicle formation and docking", whereas, in MolWorlds, the player must figure this out for themselves to reach the checkpoint. One plausible explanation for why non-native English-speakers struggled in MolWorlds was perhaps because the increased level of challenge provoked by the game mechanics, in addition to the inherent challenge of working in their non-native language, increased cognitive load [78]. However, this does not explain why native English-speakers performed poorly in SIM.

An alternative explanation is that the variable of "native language" may represent a deeper relationship between cultural identity and academic preferences/aspirations/motivations (rather than language), which may be associated with the effectiveness of educational games for culturally diverse students [79]. The design of MolSandbox, which allows for systematic exploration of molecular factors with explicit, written goals, may be more aligned with the studying habits of culturally Eastern/Asian students (who form the majority of the University's international student cohort), whereas the more playful, problem-solving quests in MolWorlds may appeal more to culturally Western students. Eastern education has been associated with rote learning and regimented studying, with less of an emphasis on creativity and problem-solving [80,81], a greater focus on the practical outcomes of education, e.g., high-level employment [82] and higher academic and career aspirations [83–85]. Societal norms around gaming may also play a role; higher play approval is typically seen amongst culturally Western parents [86,87], and may be viewed as particularly distracting or academically damaging in culturally South-Asian households [85]. This argument sup-

ports the concept that learning technologies which incorporate entertainment elements and have more of a playful, problem-solving focus may be perceived as more acceptable and effective for Western learners, whereas more structured investigation and experimentation may be more appealing for learners who identify with Eastern cultures. Survey research with 489 undergraduates at a Canadian university would suggest this is the case [88]. The authors found that the more strongly individuals identified with Canadian (western) culture, the more likely they were to have favorable views of GBL, and that individuals of South Asian heritage generally held negative views toward GBL. Unfortunately, most GBL literature (including the current publication) does not report the cultural make-up of participants, with one review finding that 67% of GBL studies did not include information regarding ethnicity [13]; this highlights the clear need for more research about the relationship between culture and the acceptability of game-based learning by students, parents, and teachers [88].

### 4.3. Moderating Effect of Gaming Habits

It was expected that frequent gamers might reconcile more misconceptions in the GBL group, since these individuals would have less of a learning curve with the game mechanics in comparison to those who rarely play video games. Prior gaming experience was, in fact, related to the number of level completions in both the GBL and SIM groups but it was not related to conceptual change. Similarly, Gauthier et al. [51] found gaming habits to be related to breadth of game completion but not to learning outcomes, and Barzilai & Blau [89] found that gaming habits and time-on-task were not related to problem-solving ability post-intervention. This indicates that the transformational potential of MolWorlds at changing misconceptions was not due to success within the game (e.g., number of levels completed, game score) but rather other factors related to the gameplay experience (i.e., PN quality, as discussed directly below).

### 4.4. Interaction Patterns Differed between GBL and SIM in Ways That Mediated Conceptual Change

It was predicted that conceptual change in the GBL and SIM groups would be directly related to their experiences whilst using the apps, specifically the presence of negative events and the frequency of correct interactions in response to the negativity (i.e., PN quality). As expected, and consistent with associated previous studies [33,34], the presence of rules and restrictions in MolWorlds resulted in significantly fewer overall interactions, fewer DCCK, and more instances of PN compared to MolSandbox. In MolWorlds, the implementation of gaming patterns (navigation and exploration, resource management, sequential level progression, and scoring) were directly responsible for increases in PN events and more strategic use of resources (i.e., % of productive DCCK), as described in pilot research [33]. Resultantly, GBL participants were more likely than SIM participants to agree to perceive the activity as difficult, and the most frequent response that MolWorlds players gave as to why they quit playing was because they were too frustrated to continue. These findings coincide with the higher PN frequency in the GBL group, but also highlights the need for good level design that balances frustration with flow, so that students remain engaged while PN ensues [90–92].

The study found that PN quality was a significant predictor of conceptual change in the GBL condition, supporting literature about the transformational power of PN in GBL [36–38]. Though MolSandbox generated higher numerical PN quality, PN quality was only associated with a reduction in misconceptions amongst GBL participants, while there was no association in the SIM group. Overall, this suggests that PN as a conceptual change strategy in interactive tools is most likely to be effective in the presence of game design, while a guarantee cannot be made in the absence of it. Past research that compares gaming environments with very similar non-gaming "control" applications support this kind of "learning guarantee" in games [34,51]. Since MolWorlds encouraged longer play time, and since 20/42 MolWorlds-players declared that they would have continued playing

had time allowed, the PN quality in the GBL group may have continued to increase, which may have led to significant differences in conceptual change between stimulus groups in a real-world scenario. Nonetheless, the interactive SIM facilitated conceptual change in non-native English-speakers, which implies that a mechanism other than PN (e.g., systematic observation, experimentation) may also be effective for some users to resolve misconceptions about molecular emergence, which requires further investigation.

*4.5. GBL Led to Longer Voluntary Engagement*

Voluntary use was significantly higher in the GBL group over the SIM group, consistent with expectations based on the literature [41–46]. Secondly, duration of use increased linearly with gaming habits in the GBL group, which supports previous research suggesting that GBL environments hold a high appeal to "gamers" [93–95]. The finding supports the suggestion that challenge—which can further a learners' sense of competency—is a primary contributor to gaming motivation [45], since more challenge was created in the GBL intervention through the addition of navigation, resource management, and reaching checkpoints. Proulx et al. [41] also suggest that movement, selection/collection, and progression (which map to the previously mentioned gaming patterns) can foster a high degree of autonomy and intrinsic motivation in learners. It also supports the suggestion that hierarchical goals sustain engagement [46]; MolWorlds had a more hierarchical goal structure than MolSandbox, including the overarching narrative goal of escaping the molecular world, level-goals of reaching each checkpoint, process-goals of initiating various cellular processes in furthering the achievement of the level-goal, and other minor goals like collecting power-ups or achieving 3-stars and a high score. Comparatively, the goals in MolSandbox were less hierarchical and essentially involved the single goal of completing the level through initiating one or more cellular processes, which may have made the user feel like they had less control [45,46], even though they had greater freedom to make modifications to the simulated environment (e.g., no power-ups needed) than GBL participants had.

Interestingly, voluntary time-on-task did not differ significantly between native language subgroups within each stimulus condition, while conceptual change did. However, differences in several self-reported engagement metrics support the finding on the relationship between conceptual change and stimulus-native-language/cultural subgroups. Native-language subgroup analysis of IMMS statements show that SIM-NES participants (who had the least conceptual change) had more negative attitudes toward MolSandbox than GBL-NES participants (who had the most conceptual change) had toward MolWorlds. Furthermore, SIM-NES participants were more likely than SIM-NNES participants to think that MolSandbox was too abstract. This may be related to why non-native English-speakers performed better than native-speakers in the SIM group; they were more perceptive to the benefit of systematic experimentation on learning, which the lack of game design in MolSandbox afforded.

*4.6. Implications and Limitations*

Overall, GBL and SIM participants achieved similar levels of conceptual change in this experiment immediately post-intervention; so, was the presence of game design in MolWorlds beneficial beyond the underlying interactive simulation? Firstly, MolWorlds players were more likely than MolSandbox users to resolve misconceptions in comparison to the baseline group, and trends revealed the GBL group were more likely to retain their conceptual change one-year later. Secondly, MolWorlds elicited longer voluntary playtimes than MolSandbox. Thirdly, the study showed that if GBL participants had experienced the same PN quality as SIM participants, they would have achieved greater conceptual change; given the time constraints of the study design, this was not the case, likely because the gameplay in MolWorlds is slower and more challenging, so the SIM intervention generated higher PN quality than the GBL intervention in the allotted time. However, PN quality was only related to conceptual change in the presence of game design and, since nearly half of

MolWorlds players said that they would have continued using the game in the same sitting if time had allowed, one might hypothesize that their PN quality and resulting conceptual change would have increased with more playtime. Therefore, it can be suggested that the addition of game design was beneficial because (1) it increased the likelihood for, and retention of, transformed concepts, (2) it increased voluntary time-on-task, and (3) it created tighter links between the learning science, interactive experiences of the players, and resulting learning outcomes. However, cultural learning preferences surrounding GBL should be considered when implementing GBL, as the approach may have varying effectiveness in an international cohort [88].

Despite these contributions, there are some limitations to the research. This study took place in a controlled setting with a limited playtime, which obscures the ecological validity of the work. An attempt was made to mediate this limitation by allowing a voluntary play context, where participants could cease playing after a given time. Future studies might consider allowing students to play at home (or in other out-of-lab contexts), but this comes with its own set of limitations around internal validity and the kinds of analyses that can be performed (e.g., qualitative analysis of screen recordings is unlikely).

Another limitation is that the baseline comparison group consisted of individuals who did not participate in the randomized controlled trial and may, based on unwillingness to volunteer, have different characteristics from those who did volunteer and were randomized to a stimulus group. This concern is somewhat mitigated by the fact that RCT participants were drawn from the broader baseline population and that the participant composition analyses showed no difference between groups in all inspected characteristics. Additionally, the sample at the one-year delayed follow-up is significantly reduced, which introduces bias in longitudinal analyses [71]. Very few non-native English-speakers were retained in the GBL group, which limits follow-up findings to native English-speakers. However, retained participants achieved similar scores on the MCAA in comparison to participants lost to attrition. Furthermore, it is unclear what other factors may have influenced students' conceptual change over the course of the year, e.g., further studies in molecular biology. This concern is partially mitigated by including the baseline control group in the analysis, who would have had the same opportunities to engage in additional studies as those in the experimental groups.

Finally, differences in conceptual change between native-language subgroups was interpreted to be related to culture, rather than language, but the author is unable to conclusively determine this without further research. Future randomized controlled trials should collect data on culture, so that its effects on GBL-related outcomes can be controlled.

## 5. Conclusions

This randomized controlled trial investigated the impact of an interactive simulation (MolSandbox) and game-based learning intervention (MolWorlds) on undergraduate students' misconceptions about molecular emergence, in comparison to a non-randomized baseline group with standard instruction. Both MolWorlds and MolSandbox helped undergraduate biology students resolve misconceptions about molecular emergence that would have otherwise remained unchanged through their undergraduate career [27]. MolSandbox afforded students the freedom to engage in systematic experimentation of various molecular factors on explicitly stated level objectives, which held high appeal for non-native English-speakers and low appeal for native English-speakers; a result that it was surmised is more closely related to differences in Western and Eastern learning cultures than it is to language itself. The value-added effect of the game design in MolWorlds was that it (1) increased the likelihood for and retention of conceptual change in comparison to the standard instruction; (2) increased voluntary time-on-task, a metric for engagement; and (3) created a play context in which in-game experiences (i.e., negativity and productive responses) have a direct impact on learning. However, caution must be taken to design game levels that balance negativity and productivity, as too much frustration may lead to

disengagement. The study highlights the effectiveness of interactive visualization design approaches for helping students resolve robust misconceptions about molecular emergence.

**Supplementary Materials:** The following supporting information can be downloaded at: https://www.mdpi.com/article/10.3390/educsci14040366/s1.

**Funding:** This work was funded by the Social Sciences and Humanities Research Council of Canada (494839), a Vanier Canada Graduate Scholarship, and by the University of Toronto's Information Technology Innovation Fund.

**Institutional Review Board Statement:** The study was conducted in accordance with the Declaration of Helsinki, and approved by the Institutional Review Board (or Ethics Committee) of the University of Toronto (protocol code 30628, 31 August 2017).

**Informed Consent Statement:** Informed consent was obtained from all subjects involved in the study.

**Data Availability Statement:** The original contributions presented in the study are included as summative data in the Supplementary Materials, further inquiries can be directed to the corresponding author.

**Acknowledgments:** The author would like to thank her former supervisor, Jodie Jenkinson, for her guidance when conducting this research. She would also like to thank the undergraduate students who participated in this research, as well as the Biology instructors who granted us access to their classrooms. Furthermore, she thanks Kara Lukasiewicz, Dani Sayeau, and other BMC master's students for their help in conducting the research, as well as the development team (Brendan Polley, Cassie Cetlin, Natalie Cormier, and Derek Ng) involved in creating MolWorlds.

**Conflicts of Interest:** The author declares no conflicts of interest. The funders had no role in the design of the study; in the collection, analyses, or interpretation of data; in the writing of the manuscript; or in the decision to publish the results.

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
