# Peer review of "Game and Simulation Stimulate Conceptual Change about Molecular Emergence in Different Ways, with Potential Cultural Implications"

_education, doi:10.3390/educsci14040366_

Round 1
Reviewer 1 Report
Comments and Suggestions for Authors
Congratulations to the author for this magnificent study. This is a case study in which different instruments are used, some widely used, and which develops a thorough and relevant analysis of the results. The conclusions fit the objectives and the discussion is sufficient. I would only like to suggest that you indicate the reliability tests applied to the data obtained, as well as the guarantees of validity of the instruments used.
Author Response
Dear reviewer,
Thank you for your kind feedback and encouragement. It is very much appreciated. Below, I outline the changes I made in the newly uploaded document based on your feedback, which I have highlighted in BLUE to make them easier to pick out.
In terms of reliability, two approaches were taken. (1) For the quantitative tests (e.g., science/bio literacy, MCAA), answers were drawn directly from a database, which removed the chance of researcher error. I have now indicated this clearly in Section 3.1. (2) For the qualitative metrics (e.g., DCCK and PN), interrater reliability was calculated using an intraclass correlation coefficient (indicated in Section 3.4.2). We achieved excellent interrater reliability in DCCK (Cronbach’s a=.967, ICC=.936, 95% CI[.853, .973]) and strong reliability in PN (Cronbach’s a=.927, ICC=.865, 95% CI[.703, .942]).
In terms of validity of the quantitative tests, I have now more clearly indicated that these were validated outside the context of the current study (Section 2.3). For example, the science and bioliteracy tests were drawn from validated work by other researchers [62], [63]. In terms of the MCAA, this was our own work which was previously validated in [5]. This reference was blinded for peer review, but it describes in detail the process of developing and iteratively testing the MCAA. I have now indicated more clearly in Section 2.3 that this is a previously validated measure, but don't want to dedicate too much extra space to this given the length of the manuscript and because, once unblinded, readers will be able to find extra details about its design and validity in the original paper.
Many thanks again for the constructive review!
Reviewer 2 Report
Comments and Suggestions for Authors
Dear Author(s),
the article is written at a high and qualitative level. The statistics is appropriately selected and used, the evaluation of the results is adequate, so I have no further suggestions for improvement, except:Disturbing are notes such as: I have written, my earlier research, etc. Avoid writing in the first person: I am, my previous research, ... write in the third person.
Author Response
Dear reviewer,
Thank you for your positive review and feedback. I have gone through the entire text and removed the first-person narrative, changing everything to third-person. These changes have been highlighted in YELLOW in the manscript to make them easy to pick out.
Thank you once again for your review.
Reviewer 3 Report
Comments and Suggestions for Authors
I think that this research lacks originality – actually the author states that ‘The aim of the current research is to confirm or refute (with a larger sample) previous findings regarding the value-added benefit of game design for facilitating conceptual change through PN about molecular emergence, in the context of robust misconceptions about molecular emergence’). I am not sure it is a good fit for this journal. Few concerns below.
32: There is no need for the self-citation since five more papers are cited.
58: There is no need for three self-citations.
159: You use this way of citing several times in text, but I think it is not correct.
187-192: I do not see why the results are contradictory.
455-457: How can you be sure that what you measure one year later is the effect of your intervention?
Comments on the Quality of English LanguageMinor editing of english language required.
Author Response
Dear reviewer,
Thank you for your constructive feedback. I respond to each of your points below, and have highlighted changes associated with your feedback in the revised document in GREEN.
Lack of originality. Indeed, the current study built on our previous pilot research using the same intervention but with a larger sample. This is not uncommon in quantitative research, where one would want to test the feasibility of the study to iron out any methodological 'kinks' before investing the time/money in a larger trial. Replication trials are also encouraged by leading authors in GBL research [14]. Furthermore, the work is appropriate for this special issue on Visualization in Biology Education because it discusses two different approaches to interactive visualization of molecular dynamics (simulation + game design) to help resolve students’ misconceptions, which previous research has shown may be perpetuated by non-interactive linear visualization (e.g., animations and illustrations).
Additionally, based on past research in GBL, the current work is original because (1) it takes a mixed-methods approach to understanding the relationship between in-game behaviours (e.g., productive negativity) and conceptual change (while this has been undertaken by the author in her previous work, it is overall an understudied area), (2) it investigates long-term retention of conceptual change, and (3) it investigates the impact of student characteristics on the effectiveness of interventions, which both have been understudied. In the introduction, I’ve now tried to highlight these aspects up front to make the novel aspects of the work clearer.
32: There is no need for the self-citation since five more papers are cited. I have removed the self-citation.
58: There is no need for three self-citations. I have removed these self-citations.
159: You use this way of citing several times in text, but I think it is not correct. I have gone through the document and corrected these. E.g., instead of “According to [15]”, I now say “According to Smith et al. [15]”.
187-192: I do not see why the results are contradictory. I have removed this wording and simply indicate the different correlations that were found between gaming habits and learning outcomes in the different studies.
455-457: How can you be sure that what you measure one year later is the effect of your intervention? I have now addressed this explicitly in the implications and limitations section 4.7. I say that it is unclear what other factors may have influenced students’ conceptual change over the course of the year, e.g., additional studies in molecular biology. However, that this concern is partially mitigated by including the baseline control group (no intervention) in the analysis, who had the same opportunities to engage in additional studies as the experimental groups.
Thank you once again for your feedback.
Round 2
Reviewer 3 Report
Comments and Suggestions for Authors
My main concern was about the originality of this paper. This concern was not addressed in the revised version. Thus, I am not sure this paper is a good fit for this journal.
Comments on the Quality of English LanguageMinor editing of english language required.